# Involvement of Acquired Tobramycin Resistance in the Shift to the Viable but Non-Culturable State in *Pseudomonas aeruginosa*

**DOI:** 10.3390/ijms241411618

**Published:** 2023-07-18

**Authors:** Gianmarco Mangiaterra, Nicholas Cedraro, Salvatore Vaiasicca, Barbara Citterio, Emanuela Frangipani, Francesca Biavasco, Carla Vignaroli

**Affiliations:** 1Department of Biomolecular Sciences, University of Urbino Carlo Bo, Via S. Chiara 27, 61029 Urbino, Italy; barbara.citterio@uniurb.it (B.C.); emanuela.frangipani@uniurb.it (E.F.); 2Department of Life and Environmental Sciences, Polytechnic University of Marche, Via Brecce Bianche, 60131 Ancona, Italy; n.cedraro@staff.univpm.it (N.C.); f.biavasco@gmail.com (F.B.); c.vignaroli@staff.univpm.it (C.V.); 3Department of Molecular and Clinical Sciences, Polytechnic University of Marche, Via Tronto 10/a, 60020 Ancona, Italy; s.vaiasicca@staff.univpm.it

**Keywords:** *Pseudomonas aeruginosa*, cystic fibrosis, antibiotic persistence, viable but non-culturable forms, tobramycin, antibiotic resistance genes, efflux pumps, qPCR

## Abstract

Persistent and viable but non-culturable (VBNC) *Pseudomonas aeruginosa* cells are mainly responsible for the recurrence and non-responsiveness to antibiotics of cystic fibrosis (CF) lung infections. The sub-inhibitory antibiotic concentrations found in the CF lung in between successive therapeutic cycles can trigger the entry into the VBNC state, albeit with a strain-specific pattern. Here, we analyzed the VBNC cell induction in the biofilms of two CF *P. aeruginosa* isolates, exposed to starvation with/without antibiotics, and investigated the putative genetic determinants involved. Total viable bacterial cells were quantified by the validated *ecfX*-targeting qPCR protocol and the VBNC cells were estimated as the difference between qPCR and cultural counts. The isolates were both subjected to whole genome sequencing, with attention focused on their carriage of aminoglycoside resistance genes and on identifying mutated toxin–antitoxin and *quorum sensing* systems. The obtained results suggest the variable contribution of different antibiotic resistance mechanisms to VBNC cell abundance, identifying a major contribution from tobramycin efflux, mediated by MexXY efflux pump overexpression. The genome analysis evidenced putative mutation hotspots, which deserve further investigation. Therefore, drug efflux could represent a crucial mechanism through which the VBNC state is entered and a potential target for anti-persistence strategies.

## 1. Introduction

Recurrence and unresponsiveness to antibiotics are two of the main features of *Pseudomonas aeruginosa* lung infections in cystic fibrosis (CF) patients [1], most likely due to the presence of bacterial subpopulations (persister cells) that are able to survive under unfavorable environmental conditions in a dormant state and to resume growth once stress factors are removed [2]. Among persisters, viable but non-culturable (VBNC) cells are unable to divide on microbiological media, but they still retain viability and, once exposed to specific stimuli, they can regain culturability. They can be considered “a deeper stage of dormancy” than persisters [3], which is induced by a wide variety of stimuli, particularly starvation. Exposure to sub-inhibitory antibiotic concentrations can also contribute to VBNC development [4], and VBNC *P. aeruginosa* has been detected in sputum samples from hospitalized CF patients undergoing antibiotic treatment [5].

We previously described the VBNC-inducing role of starvation combined with sublethal concentrations of tobramycin in an in vitro *P. aeruginosa* biofilm model [6]. In an attempt to evaluate the contributions of different mechanisms of tobramycin resistance to the development over time (three months) of the VBNC subpopulation, we started by focusing on two previously described CF *P. aeruginosa* isolates [7], one of which was characterized by acquired high-level resistance, while the other exhibited efflux-mediated low-level resistance. The obtained results suggest the major involvement of antibiotic efflux in the shift to the non-culturable state, in the presence of low-level antibiotic concentrations.

## 2. Results

### 2.1. VBNC P. aeruginosa Cell Induction in In Vitro Biofilms

The in vitro biofilms of *P. aeruginosa* PAO1-N, C30, and AR86 were developed for 48 h in lysogenic broth (LB) and then maintained for 90 days in non-nutrient (NN) broth, (i.e., M9 minimal medium lacking carbon sources), with/without tobramycin. Each strain exhibited a peculiar pattern of adaptation to the stress conditions, with different trends in VBNC subpopulation development (Figure 1).

No VBNC cells of *P. aeruginosa* PAO1-N were detected in the 48-hour biofilms developed in LB (corresponding to time 0 days in Figure 1), as the difference between the total viable cells (TVCs) and the colony-forming units (CFUs) was lower than 0.5 Log (i.e., the threshold value established to demonstrate the reliable presence of non-culturable forms [6,8]). Non-culturable forms were detected after the exposure to starvation, regardless of the presence of tobramycin, as shown by qPCR counts, which were always higher than CFU counts (Figure 1A). In particular, in the NN biofilms, a fluctuating percentage of VBNC cells was observed, which resulted in them constituting 93.6% of the TVCs after 90 days. The presence of tobramycin subinhibitory concentrations (NN + TOB) resulted, at first, in a lower amount of VBNC cells; however, these cells showed a constant increase over time, reaching and then exceeding the 90% TVCs within 90 days of antibiotic exposure, a significantly (*p* < 0.01) greater abundance.

The clinical strains C30 and AR86 both showed consistent amounts (73% and 62%, respectively) of VBNC cells in 48-hour biofilms grown in the rich medium. However, over the 90 days of stress exposure, *P. aeruginosa* AR86 exhibited a persistent population entirely made up of culturable cells (between 8 × 10^6^ and 3 × 10^7^ CFU/mL), with no VBNC cells detected after the exposure to starvation (Figure 1D).

*P. aeruginosa* C30 showed a remarkable shift to the VBNC state in the first month of starvation, which was significantly more evident (*p* < 0.01) in tobramycin-exposed biofilms (92.6% vs. 75% of TVCs). Next, the number of culturable cells gradually increased over time, representing the 100% of the whole *P. aeruginosa* population after three months (Figure 1C).

### 2.2. Evaluation of Antibiotic Susceptibility Level upon Exposure to Sublethal Tobramycin and CCCP Concentrations

The evolution over time of the tobramycin susceptibility of the *P. aeruginosa* strains C30 and AR86 under exposure to stress was evaluated by determining the minimum inhibitory concentration (MIC) and the minimum biofilm-eradicating concentration (MBEC) of tobramycin of different clones recovered from biofilm subcultures. No increases in MIC (16 and 256 µg/mL) or MBEC (256 and >2048 µg/mL) for *P. aeruginosa* C30 and AR86, respectively, were observed. In the presence of a subinhibitory concentration of the non-specific efflux-pump inhibitor carbonyl cyanide 3-chlorophenylhydrazone (CCCP), i.e., 100 µM (Appendix A), the tobramycin MIC determined in *P. aeruginosa* C30 fell to 2 µg/mL (corresponding to susceptibility), while neither *P. aeruginosa* PAO1-N norAR86 showed significant MIC variations. These results were consistent with previous observations [7], indicating antibiotic efflux as the mechanism responsible for tobramycin resistance in *P. aeruginosa* C30.

### 2.3. Ethidium Bromide Efflux and mexY Gene Expression in P. aeruginosa C30 and PAO1-N

The efflux pump activity was compared in *P. aeruginosa* PAO1-N and C30 at both the phenotypic and the gene-expression level. Firstly, the ethidium bromide (EtBr) accumulation in the bacterial cells was monitored in cultures grown in absence/presence of sub-MIC tobramycin (Figure 2).

Upon exposure to the dye, the fluorescence naturally emitted by *P. aeruginosa* increased in both strains (Figure 2A), indicating the entrance of EtBr into the cells. In *P. aeruginosa* PAO1-N, the fluorescence gradually increased, particularly in the drug-free medium, during 1.5 h of monitoring (Figure 2B, blue columns), while in *P. aeruginosa* C30, the fluorescence decreased by 30–40% compared to the starting value (Figure 2B, green columns), irrespective of the presence of tobramycin, after only 30 min of exposure. The lower fluorescence of *P. aeruginosa* PAO1-N when the strain was exposed to tobramycin is indicative of drug-induced efflux activity. By contrast, the similar rates of fluorescence decrease in both tobramycin-exposed and -unexposed *P. aeruginosa* C30 cultures suggests greater basal efflux activity. These results were supported by the gene expression analyses of the *mexY* gene, encoding for the inner protein of the MexXY-OprM system, in both *P. aeruginosa* PAO1-N and C30 (Figure 3).

Compared to the PAO1-N, *P. aeruginosa* C30 showed a greater expression of the *mexY* gene after it was grown in both the presence and the absence of tobramycin sub-MIC concentrations. This can explain the higher level of tobramycin resistance in C30 than in PAO1-N (MIC of 16 µg/mL vs. 1 µg/mL, respectively) and the greater effect of 100 µM CCCP on C30. Therefore, the inhibition of the efflux pump activity in C30 is responsible for reversion to the tobramycin susceptible phenotype.

### 2.4. P. aeruginosa Genome Analysis

To shed some light on the different levels of abundance of VBNC in *P. aeruginosa* C30 and AR86 after their exposure to tobramycin, the two strains were subjected to whole genome sequencing (WGS) and were compared in terms of their aminoglycoside resistance determinants and sequence type (ST), as well as the presence of mutations in their toxin–antitoxin (TA)/*quorum sensing* (QS) systems. The two strains belonged to different STs; specifically, *P. aeruginosa* AR86 belonged to ST260, and *P. aeruginosa* C30 belonged to ST621. Both these STs are of clinical origin, but they are not among the most difficult-to-eradicate strains in CF patients [9].

The sequencing data confirmed the carriage of the adenylyltransferase *ant(2”)-Ia* gene by *P. aeruginosa* AR86 [7] and highlighted the presence in both isolates of *aph(3′)-IIb* and *aac(6′)-Ib3*, known as aminoglycoside modifying enzymes.

Compared to *P. aeruginosa* PAO1, the two CF isolates showed several nucleotide substitutions in the TA (Appendix A) and QS (Table 1) systems. Notably, both strains harbored different mutations in the *rhlR* gene, although these did not affect the DNA-binding domain; moreover, *P. aeruginosa* C30 exhibited many substitutions in the *relB* gene coding for an antitoxin identified by the TA finder software (Appendix A).

## 3. Discussion

The culturable and VBNC persisters of *P. aeruginosa* represent a common survival strategy during recurrent and antibiotic-tolerant infections in CF patients. Although their induction upon antibiotic treatment has already been described [5], the role of sublethal antibiotic concentrations in their insurgence has been overlooked, whereas low concentrations of antibiotics can be found in the deepest layers of pulmonary biofilms between cycles of therapy [10].

We previously pointed out the prominent role of tobramycin subinhibitory concentrations in the shift of *P. aeruginosa* to the VBNC state in in vitro biofilms [6], which is putatively due to its binding to the 30S ribosomal subunit. In this study, we aimed to investigate the variability in the VBNC cell induction, after exposure to tobramycin, of *P. aeruginosa* strains showing different drug resistance mechanisms. Two previously characterized tobramycin-resistant CF isolates, i.e., *P. aeruginosa* C30 (low-level, efflux-mediated resistance) and AR86 (high-level, acquired resistance) were used [7]. Both isolates were found to be more adaptable to starvation conditions than the laboratory strain, *P. aeruginosa* PAO1-N, showing a lower decrease in culturable cells and lower amounts of VBNC cells. Potentially, the two CF strains, having already experienced the nutrient-depleted environment of the deepest layers of the lung biofilm, became more prone to growing in unfavorable conditions.

The most striking results were those obtained in the presence of the 1/4x MIC tobramycin. The laboratory strain PAO1-N shifted to the VBNC state more slowly, but with the continuous induction of VBNC forms over time, confirming previously obtained results [6]. Conversely, *P. aeruginosa* C30 exposed to tobramycin produced the greatest amount of VBNC cells (92.6%) after only 30 days of exposure, but these cells were completely lost after two further months. This could have been due either to the metabolic reactivation of the non-culturable cells [11] or to the rapid growth of the culturable subpopulation, after a first adaptation to the low drug concentrations. This increase in the culturable *P. aeruginosa* population in biofilms exposed to starvation and antibiotic pressure for extended periods is currently under consideration for future studies. Interestingly, *P. aeruginosa* AR86 never developed VBNC cells upon exposure to stress.

The bases of these three strain-specific patterns were investigated at both the phenotypic and the genetic level.

We can exclude the evolution of microbial subpopulations less affected by tobramycin than the parental strains, as no increases in MIC or MBEC values were observed in the subcultures recovered throughout the experiment.

The genome analyses evidenced relevant features regarding aminoglycoside resistance determinants and mutations in the TA and QS modules, possibly involved in the response to the drug.

Concerning acquired aminoglycoside resistance determinants, (i) the *aph(3′)-IIb* gene, mainly related to amikacin resistance, was found in both the CF strains, but also in *P. aeruginosa* PAO1 (NC_002516.2); (ii) *aac(6′)-Ib3* was found in both clinical isolates and, therefore, it was not considered critical in the strain-specific response to the tobramycin sub-MIC; (iii) the presence of the inactivating enzyme *ant(2″)-Ia* gene was found only in *P. aeruginosa* AR86 and conferred high-level resistance (MIC = 256 µg/mL); (iv) *P. aeruginosa* C30 did not harbor further aminoglycoside resistance determinants, and exhibited low-level tobramycin resistance (MIC = 16 µg/mL), which was related to the activity of chromosomal efflux pumps, probably MexXY-OprM, which is the main one involved in aminoglycoside extrusion. This assumption was experimentally verified by the EtBr-efflux and *mexY*-expression assays, which confirmed a greater activity of MexXY-OprM in this isolate compared to *P. aeruginosa* PAO1-N. Consistently, *P. aeruginosa* C30 was the only strain to exhibit increased tobramycin susceptibility when exposed to the efflux pump inhibitor, CCCP, which non-specifically hinders efflux by uncoupling the proton gradient responsible for the pumps’ activity. Considering the tobramycin-induced development of VBNC cells as a hormetic response [6], and their faster development in *P. aeruginosa* C30 compared to the laboratory strain, we can assume that the tobramycin binding to the bacterial ribosome triggers the expression of genes (still to be identified) involved in the shift to the VBNC state, similarly to the described development of tolerance to aminoglycosides [12]. The greater activity of the MexXY-OprM in *P. aeruginosa* C30 compared to *P. aeruginosa* PAO1-N after exposure to tobramycin may thus result in an early shift to the VBNC state of most of the bacterial population, followed by VBNC cell reactivation and multiplication as a consequence of drug extrusion, a behavior not observed in the absence of tobramycin. This is consistent with our previous data, which evidenced the role of the MexXY-OprM pump in the tolerance of *P. aeruginosa* to tobramycin [13] and with further studies describing efflux as a typical strategy through which this pathogen counteracts the bactericidal activities of antibiotics [14,15]. Accordingly, the lack of VBNC cell production by *P. aeruginosa* AR86 might have been due to the modification of tobramycin by Ant(2”)-Ia before reaching the intracellular environment, which prevents the action of tobramycin either as a stress factor or as a gene expression inducer.

The majority of the mutations identified in the sequences of TA and QS modules, which are known to play a crucial role in the induction of persisters [3,16,17], were shared by the two *P. aeruginosa* CF isolates and, therefore, they were not considered as critical in affecting the specific shift to the non-culturable state exhibited by each isolate. In particular, no mutations were observed in the DNA-binding domain of the *rhlR* gene, which is involved in transcription regulation. Nevertheless, the obtained results evidenced the *rhlR* (Table 1) and *relB* (Appendix A) genes as possible mutation hotspots, suggesting the appropriateness of deeper investigations on the role of specific nucleotide substitutions in these genes in the induction of the VBNC state.

## 4. Materials and Methods

### 4.1. Bacterial Strains, Growth Media, and Chemicals

The reference strains *P. aeruginosa* PAO1-N, kindly provided by Prof. Paul Williams (Centre of Biomolecular Sciences, University of Nottingham, Nottingham, UK), *P. aeruginosa* ATCC 27853, and the previously characterized CF strains *P. aeruginosa* C30 and AR86, both resistant to tobramycin, through efflux (MIC = 16 µg/mL) and aminoglycoside modification (MIC = 256 µg/mL), respectively, [7] were used. Strains were cultured in LB or cystine lactose electrolyte deficient (CLED) agar plates (all from Oxoid, Thermo Fisher Scientific, Waltham, MA, USA). Tobramycin and CCCP were purchased from Sigma-Aldrich (Saint Louis, MO, USA).

### 4.2. Antibiotic Susceptibility Test

Tobramycin MIC was determined by the broth-microdilution method, according to CLSI guidelines [18], using the concentration range of 512–0.125 µg/mL, in absence/presence of 100 µM CCCP (i.e., the amount used in ethidium bromide efflux assays to block the dye extrusion, without affecting cell viability [19]), using *P. aeruginosa* ATCC 27853 as control strain. Only discrepancies > 2 folds were considered as significant. The CCCP MIC against each strain was determined in the same way, using the concentration range of 400–12.5 µM. Tobramycin MBEC was determined according to the protocol described by Revest et al. [20], using *P. aeruginosa* biofilms grown in 96-well plates and in the concentration range of 2048–2 µg/mL.

### 4.3. VBNC-Cell Induction in P. aeruginosa Biofilms

The in vitro biofilms of *P. aeruginosa* were developed in 35 mm petri dishes in LB broth, and then sub-cultured and maintained in NN broth non-supplemented/supplemented with ¼x MIC tobramycin for 90 days, as previously described [6]. Briefly, bacterial suspensions in LB broth (OD_600_ = 0.1) were inoculated in the plates and incubated at 37 °C for 48 h, and the medium was refreshed after the first 24 h. Next, the rich LB medium was discarded and, after washing with phosphate-buffered saline, substituted with NN broth. Every 30 days, starting from the shift from the rich to the starving medium, biofilms were assessed for their content of culturable *P. aeruginosa* or TVCs.

### 4.4. Total Viable, Culturable, and VBNC Counts

Biofilms were mechanically detached and lightly sonicated; next, suitable 10-fold biofilms dilutions were used. Abundance of TVCs was determined by the previously validated *ecfX*-qPCR protocol and confirmed by *live/dead* flow-cytometry assays [6]. Culturable cells were detected by plate counts and the VBNC cell amount estimated as the difference between total viable and culturable cells.

Only discrepancies ≥0.5 Log were considered as suggestive of the presence of VBNC cells, as previously described [6,8]. All assays were performed in biological triplicate.

### 4.5. P. aeruginosa Genome Analysis

*P. aeruginosa* C30 and AR86 genomes were analyzed by WGS (MicrobesNG (https://microbesng.com/, accessed on 28 July 2021), Birmingham, UK) using the Illumina MiSeq platform and a 2 × 250 bp paired-end approach. The obtained genomes (accession numbers JAMRYN000000000 and NZ_JAGGCB000000000, respectively) were compared to the PAO1 genome (accession number NC_002516.2) using ResFinder, MLST (http://www.genomicepidemiology.org/, accessed on 28 July 2021), the Comprehensive Antibiotic Resistance Database (CARD, https://card.mcmaster.ca/, accessed on 28 July 2021) and TA (toxin-antitoxin) finder (https://bioinfo-mml.sjtu.edu.cn/TADB2/tools.html, accessed on 29 July 2021).

### 4.6. Ethidium Bromide Accumulation Assay

*P. aeruginosa* efflux pumps’ activity was measured through the ethidium bromide (EtBr) accumulation assay, performed as previously described [21]. Briefly, standardized cultures (OD_600_ = 0.1) of *P. aeruginosa* PAO1-N and C30 were washed in sterile saline solution and exposed to 2.5 µM EtBr for 1.5 h. Fluorescence emission was measured in relative fluorescence units (RFUs) by a Tecan Spark^®^ multimode microplate reader and recorded before/immediately after exposure to the dye and, subsequently, every 30 min. The signal variation was calculated as:[F_ta_ − F_t0_]/F_t0_ × 100
where F_ta_ is the fluorescence emitted at 0, 30, 60, and 90 min of EtBr exposure and F_t0_ is the fluorescence immediately after exposure to the dye. Tests were run in technical triplicates and biological duplicates.

### 4.7. mexY Gene Expression Assays

Total RNA was extracted from exponential phase cultures (OD_625_ = 0.3) in MHII broth, non-supplemented/supplemented with 1/4x MIC tobramycin, using the RNeasy Mini Kit (Qiagen) according to the manufacturer’s instructions. The RNA was reverse-transcribed (60 ng) using the QuantiTect Rev. transcription kit (Qiagen) according to the manufacturer’s recommendations. The *mexY* gene was amplified using the primers previously described [22] and data were analyzed through comparative quantitation using Qiagen’s Rotor-Gene Q MDx software. The results were expressed as percentage of gene expression in *P. aeruginosa* PAO1-N, grown without tobramycin (100%). The *rpoS* gene was used for data normalization [23]. Experiments were run in biological and technical duplicates.

### 4.8. Statistical Analysis

The significance of the differences in the VBNC amounts detected in the different experimental conditions/*P. aeruginosa* strains was assessed by Student’s *t*-test (threshold: 0.05).

## 5. Conclusions

In summary, the obtained results suggest that the ability of *P. aeruginosa* to enter the VBNC state can be influenced by the carriage of different antibiotic resistance mechanisms. Upon exposure to tobramycin, the drug’s inactivation can prevent the formation of non-culturable cells; an active efflux might instead allow a shift to the VBNC state by modulating gene expression while hindering the drug’s ability to reach its intracellular target in a sufficient amount to inhibit protein synthesis. Further studies investigating the role of specific antibiotic resistance mechanisms to the entry into persistent and VBNC states will contribute to the understanding of the onset and behavior of persistent *P. aeruginosa* infections over time and to the development of more focused antibiotic treatments.

## Figures and Tables

**Figure 1 ijms-24-11618-f001:**
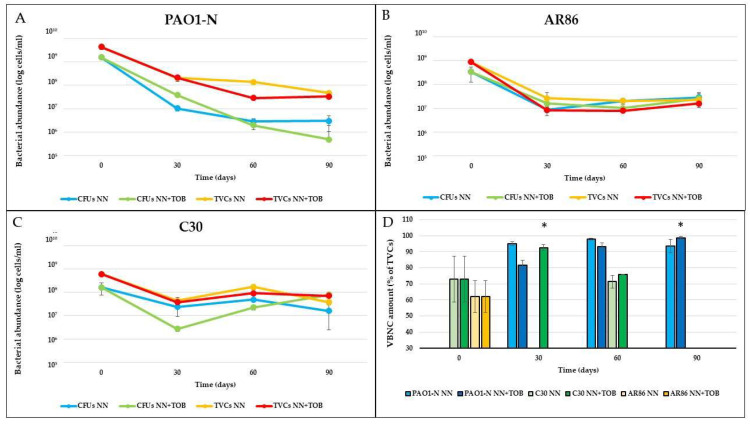
***P. aeruginosa* populations in stressed biofilms.** Culturable cells (CFUs) and total viable cells (TVCs) of *P. aeruginosa* PAO1-N (**A**), AR86 (**B**), and C30 (**C**) were quantified in starved biofilms (NN) or starved in the presence of 1/4x MIC tobramycin (NN + TOB). VBNC cell abundance, determined as difference between TVCs and CFUs, was reported as percentage of TVCs (**D**). The results are the average of three biological replicates ± standard deviation. * = *p* < 0.01.

**Figure 2 ijms-24-11618-f002:**
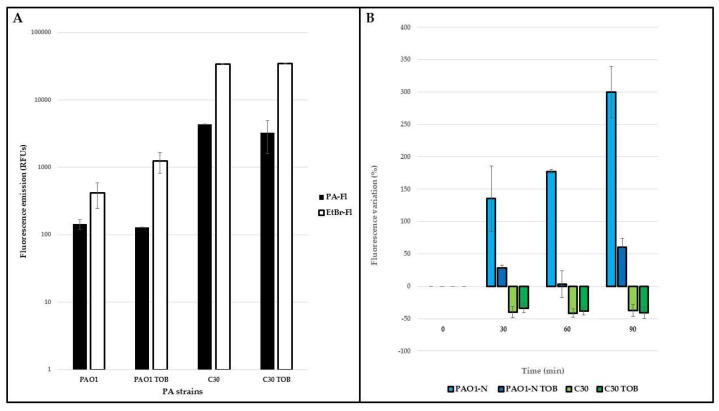
**Ethidium bromide accumulation in *P. aeruginosa***. The accumulation of ethidium bromide (EtBr) was monitored in exponential cultures of *P. aeruginosa* (PA) PAO1-N and C30, grown in absence/presence of tobramycin 1/4xMIC (TOB). Naturally emitted fluorescence (PA-Fl) and EtBr fluorescence (EtBr-Fl) emissions were measured immediately before and after exposure to the dye (**A**); subsequently, the variation of the latter was monitored every 30 min for 1.5 h (**B**). These variations are expressed as percentages (considering as 100% the fluorescence emitted by each strain immediately after exposure to EtBr) and they are reported as average of three technical and two biological replicates, ± standard deviation.

**Figure 3 ijms-24-11618-f003:**
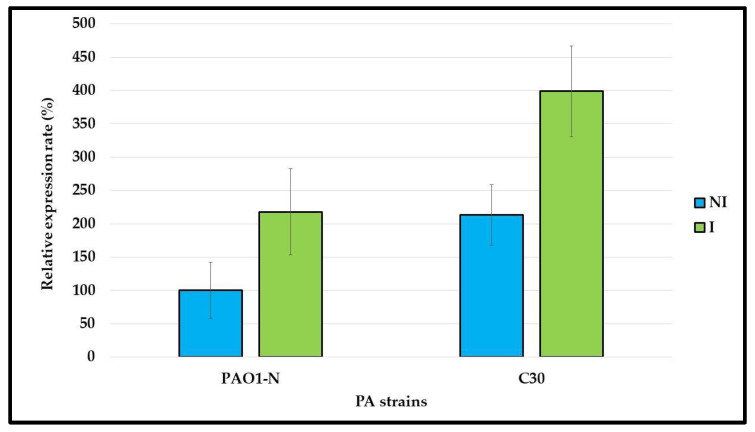
***mexY* gene expression in *P. aeruginosa*.** The *mexY* gene expression was compared in *P. aeruginosa* (PA) PAO1-N and C30, grown in absence/ presence of 1/4xMIC tobramycin, i.e., not induced (NI)/induced (I). The results, normalized by the *rpoS* gene expression, are reported as percentages of *mexY* expression in NI PAO1-N cultures, considered as 100%. Data are reported as average of technical and biological duplicates ± standard deviation.

**Table 1 ijms-24-11618-t001:** Nucleotide substitutions of *P. aeruginosa* C30 and AR86 QS regulatory genes compared to the *P. aeruginosa* PAO1 sequences.

*P. aeruginosa* Strain	Locus Tag	Gene	Nucleotide Substitutions *
C30	PA3477	*rhlR*	c.3889934A > G; c.3890096T > C; c.3890198G > T; c.3890225G > A; c.3890393G > A; **c.3890504G > A**.
AR86			c.3889934A > G; c.3890030G > A; c.3890060A > G; c.3890198G > T; c.3890418C > T; **c.3890504G > A; c.3890513G > A**.
C30	PA1430	*lasR*	c.1558525C > T.
AR86			c.1558808G > A.
C30	PA1003	*pqsR*	c.1086804G > A.
AR86			c.1086141C > T; c.1086804G > A.

* The substitutions in the autoinducer-binding domain are reported in bold.

## Data Availability

The bacterial genome sequences reported in this study can be found in the NCBI nucleotide database, accession numbers JAMRYN000000000 and NZ_JAGGCB000000000.

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
