# Peer review of "Involvement of Acquired Tobramycin Resistance in the Shift to the Viable but Non-Culturable State in Pseudomonas aeruginosa"

_ijms, 2023, doi:10.3390/ijms241411618_

Round 1
Reviewer 1 Report (Previous Reviewer 3)
Authors analyze the VBNC cell induction in biofilms of two Cystic Fibrosis Pseudomonas aeruginosa isolates, exposed to starvation with/without antibiotics, and investigate the putative genetic determinants involved. Authors suggest that while efflux pumps are likely involved in the induction of VBNC P. aeruginosa exposed to tobramycin. I do not understand how the authors reach such a conclusion. I do not understand the justification for the tests carried out. The discussion continues to be very poor. There is little discussion of results and establishment of relationships between the analyses performed.
Moreover, I disagree with the use made by the authors of the concepts persister and viable but nonculturable.
In the References Section there is disorder (capital letters, lower case,...).
Author Response
Please see the attachment.

Reviewer 2 Report (New Reviewer)
Comments:
This is a well-described study on the effect of antibiotic efflux and inactivation on the induction of the viable but non-culturable (VBNC) state in the pathogen Pseudomonas aeruginosa including in the study susceptible, tolerant and resistant strains. It is interesting the observation that drug inactivation prevents the VBNC state while drug efflux induces this phenotype. Methods, Results and conclusions were all appropriate and clearly presented.
I recommend publication of this manuscript as is, without the need for major revisions. I thank the authors for a well-written manuscript, with scope for further studies to mechanistically link the role of antibiotic resistance to the induction of VBNC - I will watch with interest!
Minor comments
Pag3 line 107. Introduce the reader to the effect of the CCCP
Pag4 line 128. Dye?
Pag5 lines 167-169. Introduce the function of the indicated genes
Pag8 line 287. Repetition of "in"
Pag9 line 333. RFU? Indicate the machine used for the fluorescence measurement.
Round 2
Reviewer 1 Report (Previous Reviewer 3)
None.
This manuscript is a resubmission of an earlier submission. The following is a list of the peer review reports and author responses from that submission.
Round 1
Reviewer 1 Report
In their work ”Acquired tobramycin resistance might influence the shift to viable but non culturable state in Pseudomonas aeruginosa”, Vignaroli et al. analyzed the induction of viable but non culturable cell in the biofilm of two Pseudomonas aeruginosa strains isolated from cystic fibrosis patients.
This study is incomplete and difficult for the reader to follow and to understand. A few examples:
· The data presented in figure 1A is in disagreement with the text in the manuscript (line 65-66). Figure 1A clearly shows the presence of VBNC cells (TVCs – CFUs) for PAO1-N. Data presented in Figure 1A is not consistent with the data presented in Figure 1D. In addition, the use of the symbols in the figures is inconsistent.
· Important/essential information and explanations are missing:
· Non-Nutrient (NN) broth is not explained (line 54). What is NN broth? This information is essential to understand and interpret the data. Is it just buffer, or are the cells still growing? This information is key to interpret and understand the data
· What is the function of e.g. rhlR? What about the other quorum sensing genes?
· CCCP is proton ionophore uncoupler and the presence of CCCP diminished/abolished the proton motive force (PMF) in bacteria. The antibiotic tobramycin belongs to the class of aminoglycosides. For bacterial uptake of tobramycin and all the other aminoglycosides an intact PMF is required, meaning Tobramycin loses its antimicrobial activity in presence of CCCP. This is in disagreement with the data (line 92-93) and conclusion presented in this study. In addition, measuring the MIC of CCCP alone is missing.
Reviewer 2 Report
The Mangiaterra et al. manuscript briefly communicates the possibility of Pseudomonas aeruginosa strains shifting to the viable but non-culturable state (VBNC). The paper is presented correctly, the methodology is described in detail and the conclusions correspond with the results.
Reviewer 3 Report
Authors in the manuscript “The Dynamic Transition of Persistence toward the Viable but Nonculturable State during Stationary Phase Is Driven by Protein Aggregation” analyze the VBNC cell induction in biofilms of two Cystic Fibrosis Pseudomonas aeruginosa isolates, exposed to starvation with/without antibiotics, and investigate the putative genetic determinants involved. Authors suggest that while efflux pumps are likely involved in the induction of VBNC P. aeruginosa exposed to tobramycin. I do not understand how the authors reach such a conclusion. The paper is interesting but the discussion is very poor and not conducting to the final conclusion.
Lines 16, 39, 123, I disagree with the use made by the authors of the concepts persister and viable but nonculturable.
Line 29. I have already indicated that it is not explained how this conclusion is reached.
Lines 39-40. Explain the concept of viable but nonculturable.
I find that they refered this article (Ayrapetyan M, Williams TC, Baxter R, Oliver JD. 2015. Viable but nonculturable and persister cells coexist stochastically and are induced by human serum. Infect Immun 83:4194–4203. doi:10.1128/IAI.00404-15.) from which I copy literal 2 definitions and 1 conclusion.
“The persister phenotype has been shown to exist stochastically within growing cultures but can also be induced by stressful environments such as starvation, oxidative stress, DNA damage, stressful pH, and antibiotics.” “VBNC cells are reported to be viable due to their intact cell membranes, low-level metabolic activity, and continued gene expression. However, they are nondividing and, unlike persisters, are unable to immediately regain the ability to divide when plated on routine laboratory medium”.
Figure 1D. Axis = percentage of VBNC cells?
Lines 74-75 and lines 79-80. From the figures presented, I do not observe this result.
Line 92. What is the function of Carbonyl Cyanide 3-Chlorophenylhydrazone (CCCP)? Why is it added?.
Line 139. Why is 1/4 x MIC tobramycin dose chosen?
Lines 144-145. “This could be either due to the metabolic reactivation of the non-144 culturable cells or to the rapid growth of the culturable cells”. Demonstration?
Lines 166-167, “In P. aeruginosa C30, the greater 166 activity of MexXY-OprM, compared to P. aeruginosa PAO1-N”. Where is analyzed?
Line 205. How are biofilms assessed?
Line 213 “Only discrepancies ≥ 0.5Log were considered as suggestive of the presence of VBNC”. Reason? Statistical analysis?
Line 233-234. Conclusion has been comented.
Round 2
Reviewer 1 Report
The authors clarified some of the points and the quality of the manuscript improved. However, the amount of data presented is limited and still incomplete, e.g. a complete table of MIC values including the MIC for CCCP alone and using a reference strain for susceptibility testing is missing. MIC values can vary from experimental setup, therefore experimental data is needed. Further studies are required to supporting the model of efflux pump being the main mechanisms for tobramycin (other efflux pump inhibitors, genetic approaches etc). The data obtained from the whole genome sequencing is interesting, but stays at a descriptive level. Overall, the observations are interesting, but the amount of data is limited and still preliminary
Author Response
The authors clarified some of the points and the quality of the manuscript improved. However, the amount of data presented is limited and still incomplete, e.g. a complete table of MIC values including the MIC for CCCP alone and using a reference strain for susceptibility testing is missing. MIC values can vary from
experimental setup, therefore experimental data is needed. Further studies are required to supporting the model of efflux pump being the main mechanisms for tobramycin (other efflux pump inhibitors, genetic approaches etc). The data obtained from the whole genome sequencing is interesting, but stays at a descriptive level. Overall, the observations are interesting, but the amount of data is limited and still preliminary.
We thank the Reviewer for the reported observations. Indeed, the aim of this Communication was to highlight the contribution of specific genes in the induction of the persistent and VBNC phenotypes. The investigation of the exact role of each gene will be issued in further studies, as part as Research Articles, to better characterize the strain-specific variability of the P. aeruginosa entrance into the dormant phenotype. However, we have included further data to answer the Reviewer’s concerns:
-We added the required Supplementary Table 1, showing the MIC values for tobramycin and CCCP against all P. aeruginosa strain, including the reference strain ATCC 27853. We have modified the manuscript accordingly (lines 98, 99, 212 and 213 of the revised manuscript).
-We demonstrated the presence of a potentiated efflux activity in P. aeruginosa C30 compared to P. aeruginosa PAO1-N using the Ethidium Bromide Agar Cartwheel method (doi: 10.2174/1874285801307010072). Using both the adopted ethidium bromide concentrations (i.e., 0.25 and 0.5 μg/ml) P. aeruginosa C30 resulted less fluorescent than the reference strain, indicating a lower concentration of the dye in the cells as a consequence of a potentiated efflux. This result is in line with the lower tobramycin MIC exhibited in presence than in absence of 100 μM CCCP and with the previous strain characterization (please see reference n 7), and further corroborates the notion of an efflux-mediated
tobramycin resistance in this specific isolate. The Ethidium Bromide Agar Cartwheel results (lines 102-105, Supplementary Figure 1) the methodological procedure (lines 216-224) and reference (#21) have been added in the revised manuscript.
As regards whole genome sequencing (WGS), we agree with the Reviewer that “the data obtained from the whole genome sequencing is interesting, but stays at a descriptive level”. This is overall true as regards results on TA and QS modules (although in agreement with literature data) in which the relationship of the found
substitutions with the VBNC state and the different behaviour of the two analysed strains is difficult to explain.
However, we would like to remark that WGS results showed the presence of the ant(2”)-Ia gene in the strain AR86 only. This finding allowed us to discuss its possible role in suppressing VBNC forms induction (lines 171, 172 and 184-186 of the revised manuscript). Further assays using a greater number of strains will provide further evidence and details about the precise molecular mechanisms of this specific response to tobramycin presence, and, therefore, they are planned for near future studies, not exceeding the extent of the present Communication.
We hope to have improved the manuscript quality and to have answered the Reviewer’s concerns. Please note that the reference list and the supplementary material have been updated.
Reviewer 3 Report
The authors have answered and/or changed most of the suggestions. However, the information given in lines 76 and 78 should be changed. 90% of VBNC cells are reached before 90 days.
Author Response
The authors have answered and/or changed most of the suggestions. However, the information given in lines 76 and 78 should be changed. 90% of VBNC cells are reached before 90 days.
We thank the Reviewer for appreciating the modifications of the manuscript to improve its quality. We have modified the sentence according to the suggestion (lines 78, 79 of the revised manuscript). We hope to have properly answered the concern raised by the Reviewer.

Round 3
Reviewer 1 Report
The authors clarified some of the points and the quality of the manuscript slightly improved. Overall, the data presented is incomplete (experiments and controls missing), speculative and preliminary (also for a communication). Some examples:
· The addition of MIC table helps the reader for understanding, but it´s e.g. unclear if the values presented in the table are before starvation or after how many days? What about the MIC values for PAO1-N in combination with CCCP? Does it change? MIC values for tobramycin in combination with CCCP (some data can be found in the text, but difficult to understand).
· Ethidium Bromide Agar Cartwheel Assay: controls without EtBr are missing – what about auto-fluorescence? Same or different for different strains? Maybe, the auto-fluorescence is higher for PAO1-N and the effect of EtBr the same after adjusting for auto-fluorescence? Most importantly, the data for AR86 is missing. Is the efflux activity the same as for C30? This is important since the authors suggest that the major difference between C30 and AR86 (line 115/116) is the presence of ant(2”)-Ia gene. It´s not clear from the text what the function of ant(2”)-Ia is, it´s called inactivating enzyme – is this specific for tobramycin? If this is specific for tobramycin, the efflux data using EtBr should be comparable between C30 and AR86?
· The authors suggest that the low-level resistance of C30 is related to the activity of chromosomal efflux pumps likely MexXY-OprM. Line 179/180: “In P. aeruginosa C30, the greater activity of MexXY-OprM, compared to P. aeruginosa PAO1-N..” This is speculation, and not supported by experimental data.